# MSTN Regulatory Network in Mongolian Horse Muscle Satellite Cells Revealed with miRNA Interference Technologies

**DOI:** 10.3390/genes13101836

**Published:** 2022-10-11

**Authors:** Undarmaa Budsuren, Tseweendolmaa Ulaangerel, Yingchao Shen, Guiqin Liu, Toli Davshilt, Minna Yi, Demuul Bold, Xinzhuang Zhang, Dongyi Bai, Dulguun Dorjgotov, Gantulga Davaakhuu, Tuyatsetseg Jambal, Bei Li, Ming Du, Manglai Dugarjav, Gerelchimeg Bou

**Affiliations:** 1Inner Mongolia Key Laboratory of Equine Genetics, Breeding and Reproduction, College of Animal Science, Inner Mongolia Agricultural University, Hohhot 010018, China; 2School of Animal Science and Biotechnology, Mongolian University of Life Sciences, Ulaanbaatar 17024, Mongolia; 3College of Agronomy and Agricultural Engineering, Liaocheng University, Liaocheng 252000, China; 4School of Industrial Technology, Mongolian University of Science and Technology, Ulaanbaatar 14191, Mongolia; 5Institute of Biology, Mongolian Academy of Science, Ulaanbaatar 13330, Mongolia

**Keywords:** horse, satellite cell, *MSTN* gene, RNA sequencing, signaling pathway, cell proliferation

## Abstract

Myostatin (*MSTN*), a member of the transforming growth factor-β superfamily, inhibits the activation of muscle satellite cells. However, the role and regulatory network of *MSTN* in equine muscle cells are not well understood yet. We discovered that *MSTN* knockdown significantly reduces the proliferation rate of equine muscle satellite cells. In addition, after the RNA sequencing of equine satellite cells transfected with *MSTN*-interference plasmid and control plasmid, an analysis of the differentially expressed genes was carried out. It was revealed that *MSTN* regulatory networks mainly involve genes related to muscle function and cell-cycle regulation, and signaling pathways, such as Notch, MAPK, and WNT. Subsequent real-time PCR in equine satellite cells and immunohistochemistry on newborn and adult muscle also verified the *MSTN* regulatory network found in RNA sequencing analysis. The results of this study provide new insight into the regulatory mechanism of equine *MSTN*.

## 1. Introduction

*MSTN* knockout increases mice skeletal muscle mass as a result of both hyperplasia (increase in number) and hypertrophy (enlargement) of the muscle fibers [1], and the mutations in the *MSTN* gene could result in the “double muscling” phenotype in cattle [2,3], sheep [4], dogs [5], and humans [6]. *MSTN*, often known as the “speed gene” in horses [7], has been acknowledged as a significant genetic factor influencing race distance aptitude [8,9]. Numerous sequence variations have been found in the *MSTN* gene’s upstream and downstream regions, and research has examined their relationships to Thoroughbred horses’ race performance [10,11,12]. Researchers have used CRISPR/Cas9 editing technologies to make *MSTN*-null horse embryos with the goal of artificially enhancing equine sports performance [13,14]. The equine’s general health is significantly impacted by skeletal muscle metabolism. The majority of research on equine skeletal muscle has concentrated on a small number of specific metabolites produced during acute exercise or training [15]. After suffering a muscle-fiber injury, skeletal muscles in particular have an extraordinary ability for regeneration, allowing for the complete restoration of their structure and function within a few weeks [16]. Direct trauma and excessive exercise can harm a horse’s myofibers over the course of a lifetime [17], and the activation [18], proliferation [19], and terminal differentiation of muscle quiescent satellite cells (SCs) [20] are crucial to their recovery. At present, little is known about the *MSTN* regulation network in equine muscle SCs, despite the fact that the impact of *MSTN* knockdown and *MSTN*-related signaling pathways in other mammalian skeletal muscle SCs have been examined [20,21,22,23,24,25]. In this study, we used miRNA interference approaches to expose the *MSTN* regulation network in equine muscle SCs because *MSTN* is an important gene for comprehending the mechanisms underpinning equine muscle development and regeneration [26].

## 2. Materials and Methods

### 2.1. In Vitro Culture of Horse Muscle SCs

Semitendinosus muscle samples were collected from healthy horses at the local slaughter house. When the muscle sample arrived at the lab, it was immediately sanitized with 70% ethanol and washed three to four times with 4-fold volume cold DPBS. After the visible adipose and connective tissues on the muscle mass were removed with a knife, all the muscles were excised and chopped into little pieces with scissors. With collagenase type IV (Sigma-Aldrich, St. Louis, MO, USA) and trypsin (0.25%, Sigma-Aldrich) solutions, fractionated enzymatic digestion was carried out for 2–30 min at 37 °C while stirring in a water bath. The cell suspension was then progressively filtered through 70 µm and 40 µm cell strainers. The cell pellet was then extracted from the filtrates using a centrifuge. The cells pellet was then transferred to culture disks and resuspended in media (20% FBS/DMEM/AB), which were then incubated at 37 °C with 5% CO_2_. In total, 1.5 h of preplating were utilized to reduce any potential fibroblast contamination. Satellite-cell-containing supernatant was then added to culture disks. Every two days, the growing media were changed. The cells were passaged at a ratio of 1:3 after reaching 80% confluence.

### 2.2. Immunofluorescence Assay

After being fixed with 4% paraformaldehyde for 30 min, the cells were rinsed with PBS containing BSA and Triton X-100, blocked for 1 h at 37 °C with PBS containing Triton X-100, and then incubated with the primary *Pax7* (AB-528428, DSHBY) and desmin (LS-B3122, LSBio) antibodies overnight at 4°C. Following three cycles of washing, samples were incubated with the secondary antibody for 1 h at room temperature. The cells were then washed three times, exposed to Hoechst (1 mg/mL, 10 min at RT), incubated, rinsed once more, and mounted.

### 2.3. Plasmid Construction and Transfection Analysis

The miRNAs targeting *MSTN* (mirRNA267 and mirRNA364) were designed with BLOCKit RNA DESIGN (https://www.thermofisher.cn/cn/zh/home/life-science/rnai.html (accessed on 7 October 2022)) based on the sequence of Mongolian horse *MSTN* CDS.

In order to create the *MSTN* knockdown plasmid, these two miRNAs with BglII and EcoRI restriction sites were separately ligated with Td Tomato-C1 vector and then transformed into the competent cells DH5α. The plasmid was verified by restriction digestion and sequencing, and finally the correct interference plasmid was purified and named Td Tomato-mirRNA267(364)-eMSTN.

SCs were transfected with the Td Tomato-miRNA plasmid using Lipofectamine 2000 (Life Science) at a plasmid density of 60,000 cells/cm^2^. At 36 h post transfection, the fluorescence signal was observed with a fluorescence microscope to verify the transfection efficiency.

### 2.4. RNA Sequencing (RNA-seq)

At 36 h, the cell culture fluid was removed and the cells were washed with PBS three times. Finally, each well was collected with 1 mL Trizol reagent (Invitrogen, Waltham, MA, USA) to collect the cell sample; then, the total RNA could be extracted immediately, or it could be stored in an ultra-low temperature refrigerator at −80 °C for later use. The RNA Library Prep Kit for Illumina (NEB, Ipswich, MA, USA) was used to create a library, and the Illumina HiseqTM2500 was used to perform the sequencing.

### 2.5. Screening of Differentially Expressed Genes (DEGs)

DEGs in *MSTN* knockdown group compared with controls of equine SCs were identified using the DESeq R package from ww.bioinfo.au.tsinghua.edu.cn/software/degseq, accessed on 7 October 2022. Differential gene screening mainly refers to the difference fold (Fold change value) and q value (padj value, corrected *p* value) as related indicators; the genes with expression |log2 fold change| ≥ 1 and *q* < 0.05 were selected as DEGs.

### 2.6. Gene Ontology (GO) and KEGG Pathway Enrichment Analysis

DEGs were annotated using the GO enrichment analysis, in which gene length bias was corrected. GO terms with the corrected value *p* < 0.05 were considered significantly enriched. Signaling pathways were investigated using KEGG (Kyoto Encyclopedia of Genes and Genomes), and KOBAS version 2.0 software. *q* < 0.05 was considered a significant value.

### 2.7. Quantitative PCR (qPCR) Assay

Trizol (Invitrogen; Thermo Fisher Scientific, Inc., Waltham, MA, USA) was used to separate the RNA from the *MSTN* knockdown and control groups, and a reverse transcription kit was used to convert the RNA samples into cDNA (Fermentas; Thermo Fisher Scientific, Inc., Pittsburgh, PA, USA). Using β-actin as a control, 14 genes were chosen for validation of the RNA sequencing gene profiling results. In total, 10 µL of SYBR Green PCR Master Mix, 2 µL of cDNA, 1.2 µL of each primer (10 µM), and 6.8 µL of RNase-free water made up the SYBR Green PCR experiment. The cycling schedules were as follows: 95 °C for 1 min, 95 °C for 15 s, 60 °C for 30 s, and 72 °C for 30 s. Using the 2^−∆∆CT^ approach, the levels of gene expression were calculated. The parallel experiment has at least three replicates, and the data’s mean value of Mean ± S.D. is shown. In this study, a significance level of *p* < 0.05 is accepted. Appendix A contains a list of the primers utilized for this study.

### 2.8. Immunohistochemistry

Equine muscle cryosections of 8 µm were prepared from the Tissue-Tek embedded samples and were collected onto coated glass slides (Thermo Scientific SuperFrost Plus Adhesion slides, Fisher scientific, Brussels, Belgium) and stored at −20 °C. The muscle cryosections were air-dried and then blocked for 120 min in 5% defat milk in PBS solution at room temperature. The slides were washed with permeabilization solution (BSA and 0.2% triton in PBS solution) and incubated overnight at −4 °C with primary antibodies for *MSTN* (DF13273, Affinity, Hong Kong, China), *MyoD1* (MA5-12902, Thermo Fisher), and *MyoG* (A17427, ABclonal, Wuhan, China) at a dilution of 1:500. After the slides were rinsed 5 min in PBS, they were washed briefly with permeabilization solution and incubated with the secondary antibodies dissolved in 0.5% BSA in PBS for 1 h at room temperature. Subsequently, after rinsing the slides for 5 min in PBS, the DAB Horseradish Peroxidase Color Development Kit (HZ-0010, Luzhen Biology, China) was applied. The staining intensity of 10 random units at a 200× amplification (with each muscle fascicle as the measure unit) was quantified by the well-established Image-J-software-based method indicated in the previous studies [27,28,29].

### 2.9. Statistics

There are at least three replicates of the parallel experiment, and the mean value of Mean ± S.D is indicated for the data. All data were analyzed by SPSS16.0 using two-way ANOVA with repeated measures, followed by Tukey’s test. A difference was deemed statistically significant if the *p*-value was less than 0.05.

## 3. Results

### 3.1. The Characteristics of Horse Muscle SCs

We were able to successfully separate and purify the Mongolian domestic horse muscle satellite cells (HMSC) in vitro using the methods of enzyme digestion and differential adhesion. The horse fetal fibroblast (HFF) is a polygonal cell type, as can be seen in (Figure 1A), whereas the HMSC have finer edges and are primarily spindle-shaped. Additionally, HMSC demonstrated a different rate of in vitro proliferation, demonstrating that its proliferation is slower than that of HFF (*p* < 0.05). (Figure 1B). Additionally, the results of qPCR (Figure 1C) and cell immunofluorescence test (Figure 1D–F) jointly demonstrated that HMSC, but not HFF, exhibits significant expression of genes specific to mammalian SCs, such as *Desmin*, *Pax7*, and *MyoD1*.

### 3.2. MSTN Knockdown Accelerates HMSC Cell Growth

The interference plasmids miR267 and miR364 were created in this investigation by designing two miRNAs (miR267, miR364) that target horse *MSTN* mRNA and cloning them into the Td Tomato-C1 plasmid (Figure 2A). The CtrlmiR plasmid containing a control miRNA was employed in this investigation as a control. Based on the red fluorescence of Td Tomato, the transfection efficiencies of three different plasmid types reached 80% after 24 h (Figure 2B), and the expression of *MSTN* mRNA was dramatically repressed (*p* < 0.05) (Figure 2C). Furthermore, we discovered that *MSTN* knockdown might greatly improve the cell proliferation rate of HMSC during subsequent culture, particularly during the first 48 to 72 h (Figure 2D).

### 3.3. MSTN Knockdown Alters the Transcriptome

Two samples of each cell line were sequenced, and RNA-seq libraries were created for the control (transfected with CtrlmiR) and experimental (transfected with miR267 and 364) groups to profile gene expression after MSTN knockdown. After eliminating adapters and removing low-quality reads, the sequencing provided a high-quality dataset (Figure 3A–C) and about 78,134,605 clean reads.

### 3.4. DEG Analysis Reveals the MSTN Regulatory Networks in HSMCs

To examine variations in gene expression, read count data from the transcriptome were used. There were 598 DEGs in all that showed a difference expression in the experimental and control groups. When the threshold values were *q* < 0.05 and |log2 Fold change| ≥ 1, 427 of them were up-regulated and 171 were down-regulated in the experimental groups compared to the control group (Figure 3D,E). Three categories were created from the results of the GO enrichment study of DEGs: biological process (BP), cellular component (CC), and molecular function (MF). The cellular activities, developmental processes, and biological regulation in terms of BP; the cell part, organelles, and membranes in terms of CC; and the catalytic activity, signal transduction activity, and binding in terms of MF were shown to be enriched in the down- and up-regulated genes (Figure 3F). According to KEGG analysis, DEGs are involved in 37 pathways, including the mitogen-activated protein kinase (MAPK) signaling pathway, the TGF-β signaling pathway, the signaling pathway that controls stem cell pluripotency, the signaling pathway that controls the actin skeleton, the PI3A-Akt signaling pathways, the cancer pathways, and the signaling pathway for cell adhesion molecules (Figure 3G). In total, 14 DEGs associated with myogenesis and the satellite cell cycle were chosen for qPCR experiments to confirm the RNA-seq results. Overall, the qPCR and RNA-seq results were well correlated, proving the accuracy and reliability of the RNA-seq data (Figure 3H).

### 3.5. The MSTN Regulatory Network in the Horse Muscle Was Verified by Immunohistochemistry Data

We used immunohistochemistry to examine the expression of *MSTN*, *MyoG*, and *MyoD1* in adult and newborn equine muscle tissues because the in vivo expression of *MSTN* in muscle tissue declines with age. This allowed us to determine whether the expression of the positively regulated DEG *MyoG*, and the negatively regulated DEG *MyoD1* also changed in a manner that was consistent with age. The immunohistochemistry results confirmed the findings of RNA-seq and qPCR, demonstrating that *MSTN* negatively regulates *MyoD1* expression while positively regulating *MyoG* at the protein level (Figure 4).

## 4. Discussion

An ancient horse breed known as the Mongolian horse, which has long been a staple of the nomadic pastoral herders’ culture in North Asia [30,31], is renowned for its exceptional endurance and robust genetic variety [30,32]. As a result, the Inner Mongolia Autonomous Region of China and Mongolia consider the Mongolian horse to be one of the most significant breeds of traditional long-distance races [32].

The procedure of harvesting SCs from equine muscles was first described in 1992 [33] and has been used to examine the impact of exercise on muscle tissue [34]. However, there haven’t been many investigations into the processes that underlie *MSTN*’s function in the horse model. When tissue is damaged or injured, SCs change to a proliferative state, which enables the production of a large cell pool suited for myogenic differentiation [35]. Our research shows that, similar to SCs from other species, SCs from horses exclusively express *Pax7*, *MyoD1*, and *Desmin* [36,37,38]. In the population of adult SCs, *Pax7* is expressed in both quiescent and active conditions [39]. Equine SCs grown in culture have “wedge” morphologies and have condensed interphase chromatin, which is in line with the idea that the majority of SCs in resting muscles are quiescent and transcriptionally inactive [40,41].

The *MSTN* CDS region’s full-length sequence, which was cloned from a muscle sample from a Mongolian horse in the current work, is 1134 bp, and it is entirely consistent with the Thoroughbred *MSTN* CDS sequence supplied by NCBI (Accession number: NM_001081817.1). We discovered that *MSTN* knockdown could boost the SCs’ proliferation rate through the very effective RNAi studies. The *MSTN* knockdown investigations on the SCs of other animals likewise reported the same behavior. A previous mouse study discovered that *MSTN* functions through a complex regulatory network that includes PAX, Myosin family proteins, WNT, MAPKmTOR family members, and numerous genes related to *CDKN1C (p57, Kip2)* that control the cell cycle [42]. Similar to this, our RNA-seq analysis following *MSTN* knockdown revealed that *MSTN* is involved in the regulation of multiple pathways crucial for the activities of muscle cells in horse muscle SCs.

According to our findings, we discovered that a number of DEGs are connected to the PI3K/AKT/mTOR signaling pathways, which are crucial for mediating a variety of cellular functions, including nutrition uptake, anabolic responses, cell development, and survival. Phosphatidylinositol 3-kinase (PI3Ks), AKT, and the mammalian target of rapamycin (mTOR, also known as mechanistic TOR), which make up the heart of this pathway, are frequently over-activated in most malignancies and have thus come to be the subject of research in this area [43]. Additionally, it has been noted that the *MSTN*-Smad pathway affects protein kinase AKT’s activity, preventing the mTOR pathway and protein synthesis [44]. DEGs belonging to WNT signaling also emphasize the critical functions of WNT members in muscles. The previous study found that WNT signaling might be activated after muscle damage and that myogenic cells’ TCF reporter activity increased two days after muscle damage [45]. Unlike *Wnt1*, *Wnt3a*, and *Wnt5a*, which promote SCs proliferation, *Wnt4* and *Wnt6* inhibit it [19]. Although Wnt4 has a beneficial role in controlling the proliferation of SCs, *MSTN* can operate as *Wnt4*’s upstream antagonist to prevent *Wnt4*-mediated SCs growth. Additionally, it has been demonstrated that *MSTN* increases the expression of the WNT signaling pathway inhibitors *sFRP1* and *sFRP2* [46]. We discovered that *Wnt5a* greatly increases following *MSTN* interference, suggesting that WNT is also involved in the SCs regulation controlled by *MSTN*. We also focused on aspects of cell cycle regulation in this work. SCs are in a quiescent state in the adult resting muscle [47], which is characterized by a low rate of metabolism, a lack of cell cycling (G_0_ phase), and low RNA concentration [48]. In cell-culture studies, mechanistically, *MSTN* interacts with the cell cycle machinery to induce the cell cycle exit during the gap phases (G_1_ and G_2_) [49]. In bovine skeletal muscle SCs, *MSTN* knockdown resulted in an increase in *CDK2* expression, a decrease in *P21* expression, and a stimulation of proliferation [50]. The increased production of the cyclin-dependent kinase inhibitor, *P21*, which further suppresses the gene essential for the G_1_/S transition, *Cdk2*, is thought to be the cause of the *MSTN*-triggered halt of the cell cycle and the inhibition of proliferation [51,52,53]. Compared with activated/cycling SCs, many negative regulators of the cell cycle and myogenic inhibitors, including cyclin-dependent kinase inhibitors 1B (*Cdkn1b*; also known as *p27* or *p27Kip1* [54]) and 1C (*Cdkn1c*, also known as *p57* or *p57Kip2*); Rb [55] (also known as *Rb1*); and *Rgs2*, *Rgs5*, *Pmp22*, and FGF suppressor gene *Spry1* [56], were highly up-regulated in quiescent SCs [57,58]. Our RNA-seq results mainly agree with those from studies on different species and models, suggesting that one of the main roles of *MSTN* is cell-cycle regulation.

*Pax7* is robustly and consistently expressed in the SCs of adult muscle, whereas *Pax3* is often expressed at very low levels, with the exception of muscles such as the diaphragm [59]. In our study, *Pax7* and *Pax3* expression did not alter significantly; however, *Pax5* expression significantly decreased. This distinction leads us to believe that different species have different regulatory roles for PAX family proteins in the SCs. However, more research is required to make this clear. Myogenic cells can selectively express the genes *Myf5* and *MyoD1*, whereas resting SCs cannot. When SCs enter the cell cycle again and differentiate into highly proliferating myoblasts, *Myf5* and *MyoD1* are elevated. Some cells exit the cell cycle during the proliferation phase, start to express *MyoG*, and then differentiate into new muscle fibers, while other cells lose their myogenic capabilities by lowering the level of *MyoD1* expression and return to a resting state to refill the SCs pool [60]. In our research, after *MSTN* knockdown in horse muscle SCs, *MyoD1* increased, whereas *MyoG* decreased. Together with the age-related *MSTN* downregulation-induced *MyoD1* increase and *MyoG* decrease in horse muscle, the critical role of *MSTN* in the entire life cycle of muscle cells is highlighted here.

In conclusion, the *MSTN* regulatory network described here advances our understanding of the function and activity of equine muscle SCs and encourages the use of this information in tissue engineering, regenerative medicine, and related fields of study.

## Figures and Tables

**Figure 1 genes-13-01836-f001:**
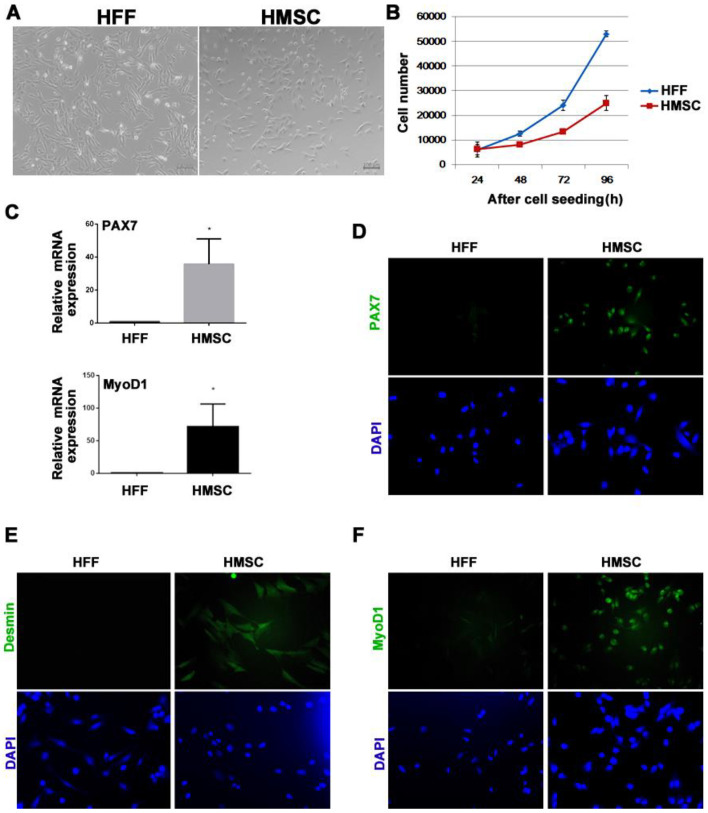
HMSC characterization. (**A**) Morphology of HFF and HMSC in vitro. (**B**) Proliferation curves (*p* < 0.05) of HFF (blue line) and HMSC (red line). (**C**) qPCR determination of *Pax7* and *MyoD1* mRNA expression in HFF and HMSC * *p* < 0.05. (**D**–**F**) Immunofluorescence results of *Pax7*, *Desmin*, and *MyoD1* in HFF and HMSC.

**Figure 2 genes-13-01836-f002:**
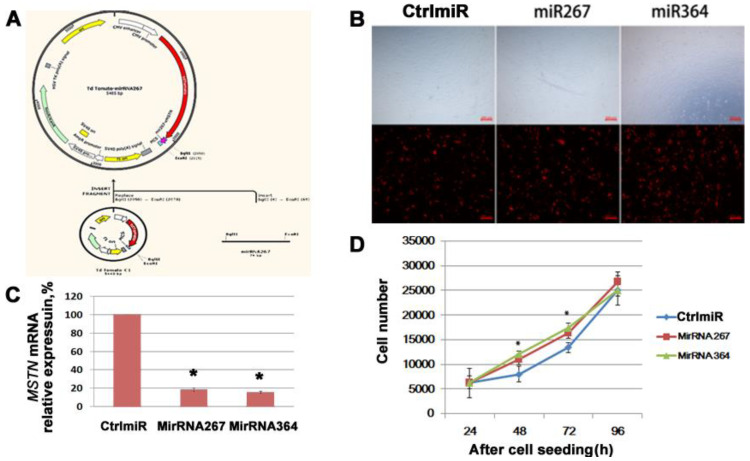
The effect of *MSTN* knockdown on HMSC proliferation. (**A**) Plasmid construction images in SnapGeneTM1.1.3 Software for equine *MSTN* knockdown plasmid. (**B**) After 24h transfection in Mongolian horse SCs in monolayer cultures. (**C**) qPCR validation of the interference efficiency. (**D**) Cell proliferation curves for different groups. * represents *p* < 0.05.

**Figure 3 genes-13-01836-f003:**
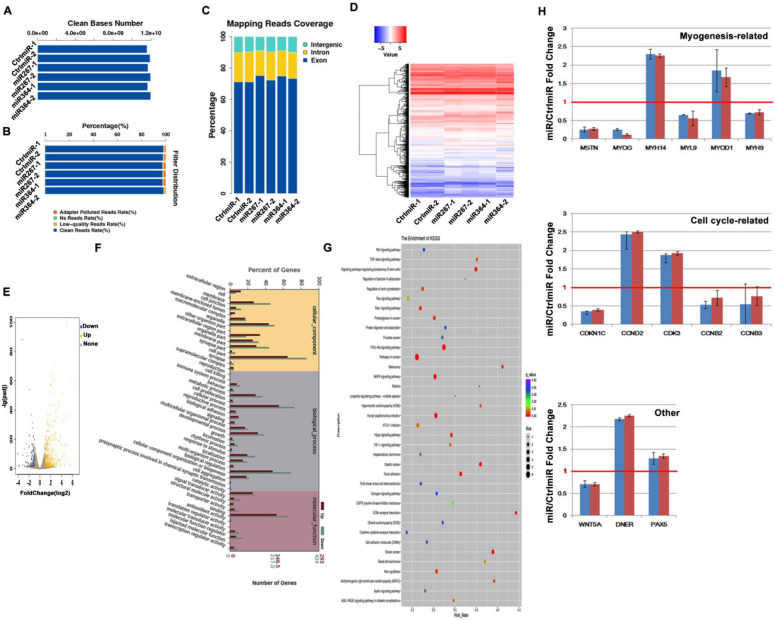
Analysis of differentially expressed genes (DEGs) after *MSTN* knockdown in HMSC. (**A**) Clean base number of all sample readings. (**B**) Filtering distribution map indicates the high quality of readings. (**C**) The location of readings indicates the normal distribution ratio of read sequences. (**D**) Clustering of DEGs among samples. (**E**) Volcano plot for the DEGs between control group and *MSTN* knockdown group. (**F**) Histogram of GO enrichment of DEGs. (**G**) KEGG pathway analysis on DEGs. (**H**) qPCR validation of DEGs. Red line represents the gene expression level in control group, and all bars represent the expression levels of tested genes in experimental groups treated with different miRNA targeting *MSTN*. All tested genes in experimental groups showed significant changes at *p* < 0.05 compared to the control group.

**Figure 4 genes-13-01836-f004:**
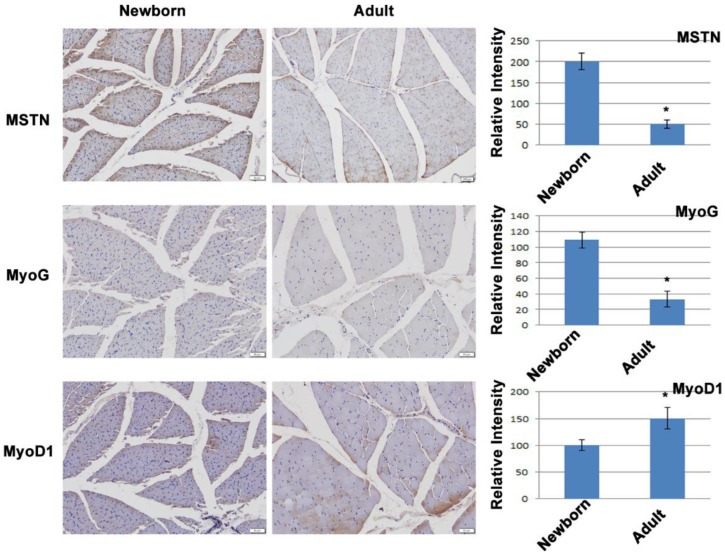
Immunohistochemistry staining of *MSTN*, *MyoG*, and *MyoD1* expression in newborn and adult equine muscles. The positive signals are quantified in left bar charts. * Indicates the significance at the level of *p* < 0.05.

## Data Availability

The datasets used and/or analyzed during the current study are available from the corresponding author on reasonable request.

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
