# Peer review of "MSTN Regulatory Network in Mongolian Horse Muscle Satellite Cells Revealed with miRNA Interference Technologies"

_genes, 2022, doi:10.3390/genes13101836_

Round 1
Reviewer 1 Report
By using RNA interference-mediated MSTN silencing, Undarmaa Budsuren et al reported that MSTN silencing significantly decreases 22 equine muscle satellite cell proliferation rate. They further validated MSTN regulatory network in equine satellite cells mainly involves genes related to muscle function (PAX and myolysis factor family) and cell cycle regulation, and signaling pathways, such as Notch, MAPK, and WNT. The project is meaningful and the findings in the paper have great guiding significance for practical application. Some minor concerns should be addressed;
(1). The language should be revised. There are several obvious spelling or grammar mistakes, for example: in line 21, “RNA interference-mediated by microRNAs”; line 51, “DMEN/F12 medium”.
(2). The figure resolution should be improved. It’s hard to see clearly for some panels. For example Fig 2A. It’s hard to know the details of each element on the vector.
(3). The discussion part is too long. It would be better if the authors could move the part of the discussion into the introduction part.
Reviewer 2 Report
The authors used RNAseq to reveal MSTN regulatory in Mongolian horse muscle satellite cells. The dataset is an important resource for the regulation of MSTN in horse muscle regeneration and satellite cell proliferation and differentiation. However, the major concern is that MSTN expressions from young and adult horse are a separated story from the main story. Here are the detailed suggestions:
Line 51: DMEN/F12 means DMEM/F12?
Line 68: Please briefly describe how to use the primary explant technique to isolate satellite cell or cite the published paper before.
Line 166-169 No qPCR results showed in Figure 4.
Line 199: Can you please check the Pax7, Myod1 and Desmin protein expression after MSTN knockdown using immunofluorescence staining or western blot? The protein level is more important than the RNA level.
Figure 2 C: For growth curve during cell culture, I suggest that the MTT assay is more convincing than count the cells.
Figure 3 E: Please use Venn diagram or volcano plot to show the DEGs.
Figure 3 H: Please mark the significant difference in expression levels of key genes.
Figure 4: For the MSTN image of adult equine, please replace the one with clear hematoxylin staining. As MSTN is a secreted ligand of the TGF-beta, but Myod1 and MyoG are transcriptional factors located in the nucleus, the qualification way should be different. Please add the qualification assay in the Materials and Methods and explain how to quality (MSTN) a secreted protein.
Line 242 and Figure 4: For the results of MSTN expressions from young and adult horse, it seems it is a separated story from the main story, and this result even did not show in your abstract. Can you please find the cues from your bulk RNAseq that linked to the result, and adding the MSTN expressions from young and adult mouse, human or etc. on introduction and discussion?
Round 2
Reviewer 2 Report
Line 68: Please briefly describe how to use the primary explant technique to isolate satellite cell or cite the published paper before.
A: Thank you for your suggestions. To further characterize the effect of MSTN knockdown on equine model, the primary explant technique was used for experimental two groups, which are satellite cell identification and plasmid transfection analysis. Briefly, the growth medium was changed every 2 days and after obtaining 80% confluence, the cells were passaged at a ratio of 1:3. After passing to the second generation, some cells were cryopreserved in liquid nitrogen with cryopreservation solution (10% DMSO + 90% FBS), some cells were used for transfection experiments. Please see line 76-79.
Follow-up: I think we are more interested in how you get satellite cell for horse muscle. For example, used blaze to mince the muscle tissue and followed by 800 U collagenase II and 1.1 U dispase to get the single cell suspension, and do FACS sorting to get Vacam1+; Sca1-; CD31-;CD45- population (doi:10.1038/nprot.2015.110). Therefore, please explain the primary explant technique in your Materials and Methods.
Line 166-169 No qPCR results showed in Figure 4
A: Thank you for your comments. The title of qPCR results has been added in image description of Figure 4 accordingly. Please see line 248.
Follow-up: As this paragraph describes the method of histochemistry of equine muscle tissue, the sentences of “To validate the results of 164 histochemistry, realtime PCR analysis was used to detect the expression levels of the of the related genes. RNA was extracted from muscle samples and reversely transcribed into cDNA using a reverse transcription kit. The relative mRNA abundance was normalized with β-actin gene using the 2−ΔΔCt method.” are improper here, please move to line 149.
Line 242 and Figure 4: For the results of MSTN expressions from young and adult horse, it seems it is a separated story from the main story, and this result even did not show in your abstract. Can you please find the cues from your bulk RNAseq that linked to the result, and adding the MSTN expressions from young and adult mouse, human or etc. on introduction and discussion?
A: Thank you very much for your suggestions. This part of result is the logical continuation of our RNA seq. Because RNAseq showed us the correlation between main regulators MSTN and Myod1/MyoG. Through the histochemistry result, we confirmed that when MSTN expression decreases with the age, the Myod1/MyoG expression showed an exactly same trends indicated in RNAseq after MSTN downregulation. In vivo data proved the validity of our in vitro study on equine satellite cells.
Follow-up: please add this result to your abstract. And please add the connection between main regulators MSTN and age to the Discussion part.
